# Challenging the highstand-dormant paradigm for land-detached submarine canyons

M. S. Heijnen[1], F. Mienis[2], A. R. Gates [1], B. J. Bett [1], R. A. Hall [3], J. Hunt [1], I. A. Kane [4], C. Pebody[1], V. A. I. Huvenne [1], E. L. Soutter[4] & M. A. Clare [1]✉

Sediment, nutrients, organic carbon and pollutants are funnelled down submarine canyons from continental shelves by sediment-laden flows called turbidity currents, which dominate particulate transfer to the deep sea. Post-glacial sea-level rise disconnected more than three quarters of the >9000 submarine canyons worldwide from their former river or long-shore drift sediment inputs. Existing models therefore assume that land-detached submarine canyons are dormant in the present-day; however, monitoring has focused on land-attached canyons and this paradigm remains untested. Here we present the most detailed field measurements yet of turbidity currents within a land-detached submarine canyon, documenting a remarkably similar frequency (6 yr$^{-1}$) and speed (up to 5–8 ms$^{-1}$) to those in large land-attached submarine canyons. Major triggers such as storms or earthquakes are not required; instead, seasonal variations in cross-shelf sediment transport explain temporal-clustering of flows, and why the storm season is surprisingly absent of turbidity currents. As >1000 other canyons have a similar configuration, we propose that contemporary deep-sea particulate transport via such land-detached canyons may have been dramatically under-estimated.

[1] National Oceanography Centre, European Way, Southampton, UK. [2] Department of Ocean Systems, Royal Netherlands Institute for Sea Research (NIOZ-Texel), Den Burg, The Netherlands. [3] Centre for Ocean and Atmospheric Sciences, School of Environmental Sciences, University of East Anglia, Norwich Research Park, Norwich, UK. [4] Department of Earth and Environmental Sciences, University of Manchester, Manchester, UK. ✉email: michael.clare@noc.ac.uk

Submarine canyons are found on all the world's submerged continental margins, often dwarfing onshore river systems[1,2]. These incised conduits provide the dominant connection for sediment, nutrients and pollutants from continental shelves to the deep sea, enhance primary productivity, and locally steer ocean currents[3–11]. The resultant biodiversity hotspots underpin diverse and important ocean ecosystems[12–14]. Turbidity currents that flow along submarine canyons can travel 1000s of km, transporting more sediment than rivers on land, and efficiently burying large quantities of organic carbon, thus contributing to regulation of climate on geological timescales[15–17]. The fast and dense nature of these currents also poses a threat to the network of seafloor cables that underpins the internet and global communications[18,19]. It is therefore important to understand the frequency, magnitude and controls on turbidity current activity and how this varies between submarine canyons worldwide.

The current paradigm holds that turbidity current activity should vary between different canyon types, as a function of their physiography and relative sea level[20–23]. Where canyons cut sufficiently far landward into the continental shelf (land-attached), they maintain direct connection with fluvial or long-shore sediment supplies on the inner shelf during both sea-level high- and lowstands[11]. For instance, the Congo Canyon (W Africa) extends into the estuary to provide direct connection to the high-discharge Congo River across all sea-level stands[23,24]. The head of Monterey Canyon (California) lies only a few metres from shore, intersecting two littoral cells that directly funnel sediment into the canyon head[25,26]. In contrast, land-detached canyons that do not, or barely, incise into the continental shelf are generally considered to be inactive during sea level highstands, only switching on during lowstands[11,23]. This issue is particularly compounded for broad continental shelves, which were extensively flooded following the Last Glacial Maximum. Examples include the eastern margin of N and S America, W Australia, the Arctic, and submarine canyons of the Celtic Margin, NE Atlantic, that are disconnected by 100s of km from previous lowstand fluvial inputs or littoral cells (which are limited to <5 km from the shore)[1,11,13,27–30]. Therefore, land-detached systems such as the Whittard Canyon on the Celtic Margin, whose head lies 300 km from the nearest coastline (Fig. 1), should be dormant with respect to turbidity currents in present-day highstand conditions.

Advances in technology have enabled the direct measurement of turbidity currents, providing the first field-scale observations of their triggering, down-slope evolution and internal structure (e.g[17,19,26,31].). These studies focused on submarine canyons that connect directly with major sediment supplies (either river- or littorally fed), recording frequent (sub-annual) turbidity currents, with velocities of ~1–8 ms⁻¹ [17,19,26]. This bias towards the monitoring of land-attached canyons exposes a major gap in our understanding of land-detached submarine canyons, which account for 81% of large submarine canyons worldwide[11], and their controls on global deep-sea particulate transport.

Here we report the most detailed direct measurements yet of turbidity currents in a land-detached submarine canyon to answer the following questions. First, how does the activity of turbidity currents in a land-detached submarine canyon compare to land-attached systems? Rather than being dormant, we show, for the first time, that powerful (up to 5–8 ms⁻¹) and frequent turbidity currents operate within Whittard Canyon, which are surprisingly similar to the measured recurrence and velocities in major land-attached canyons[17,19]. Second, what causes these turbidity currents and can they be predicted? Monitoring studies in land-attached submarine canyons have shown exceptional external events (e.g. earthquakes, floods, storms) are not necessarily needed to trigger powerful flows[26,32]. We investigate

whether this is similarly the case for land-detached systems and which processes are responsible. Finally, if powerful and frequent turbidity currents can occur within Whittard Canyon, what are the implications for the many other similar land-detached canyons worldwide and for contemporary transfer of sediment, carbon and pollutants to the deep sea?

We answer these questions by analysing direct flow monitoring data acquired using Acoustic Doppler Current Profilers (ADCPs) on two deep-water moorings in the eastern branch of Whittard Canyon, which recorded profiles of water column velocity and backscatter (a proxy for sediment concentration[17,19]) from June 2019 to August 2020 (Fig. 1). A down-looking high-frequency (600 kHz) ADCP positioned 30 m above the seafloor on Mooring M1 (1591 m water depth; 26 km downstream of the canyon head) recorded at 1 m intervals, every 5 min. Mooring M2 (2259 m water depth; 47 km downstream of the canyon head) included a 75 kHz up-looking ADCP placed 14 m above seafloor that recorded at 16 m intervals, every hour. The ADCP at M2 operated with a blanking distance of 24 m, thus inhibiting detection of flows <39 m thick. A sediment trap was also located at 10 m above the seafloor on Mooring M1 to sample suspended sediment. This trap was programmed to sample for 18 days, after which point a carousel mechanism rotated a new sampling bottle into place. Two surface buoys managed by the UK MetOffice provided hourly meteorological data during the monitoring period (Figs. 1 and 2).

## Results and discussion

**Surprising turbidity current activity in a land-detached canyon.** Six turbidity currents were detected within the 1-year monitoring window, with maximum ADCP-measured velocities of 1.1–5.0 ms⁻¹ (Figs. 2 and 3; Table 1). All six flows were recorded at M1, three of which were also clearly identified 21 km downstream at M2. The two slowest (1.1–1.3 ms⁻¹) and thinnest (15–20 m) flows (Flows 3 & 4) were not detected at M2; either dissipating before reaching M2, or were too thin for detection by the lower frequency upward-looking ADCP. It is not possible to discern whether Flow 6 reached M2 as it occurred shortly after M2 stopped recording (Fig. 2). Transit velocities of the turbidity currents estimated between M1 and M2 range from 2.7 to 8.0 ms⁻¹ (Table 1). The instantaneous (ADCP-recorded) velocities recorded at M1 are similar to the transit velocities of the respective flows, albeit at the lower end; in keeping with previous results that indicate ADCPs often under-record velocities compared to transit speeds[19,33]. Upon recovery of M1, the first sediment trap bottle and overlying funnel were completely filled with well-sorted sediment with an average grain size of 121 μm, and a maximum of 460 μm (Fig. S2). This significant sedimentation event occurred within the first 18 days of monitoring; a time window that included the first turbidity current, hence we interpret that this, and presumably later flows, were capable of suspending fine sand at a height of at least 10 m above the seafloor. Sediment stocks, and hence the source of the flows, likely derive from extensive sediment wave fields at the Whittard Canyon rim that dominantly comprise fine to coarse sand, for which off-shelf transport has been inferred from high-resolution seafloor surveys[34]. Plastic fishing line was observed wrapped around the M1 and M2 mooring anchor chains (Fig. S1), supporting evidence of active transport of litter as inferred from previous studies[13].

**A frequency and magnitude of flows equivalent to active land-attached canyons.** Dating of cored deposits previously indicated that episodic turbidity currents reached the Celtic Fan at the distal end of the Whittard Canyon within at least the last 2,000 years, and turbidity current deposits accumulated in the proximal

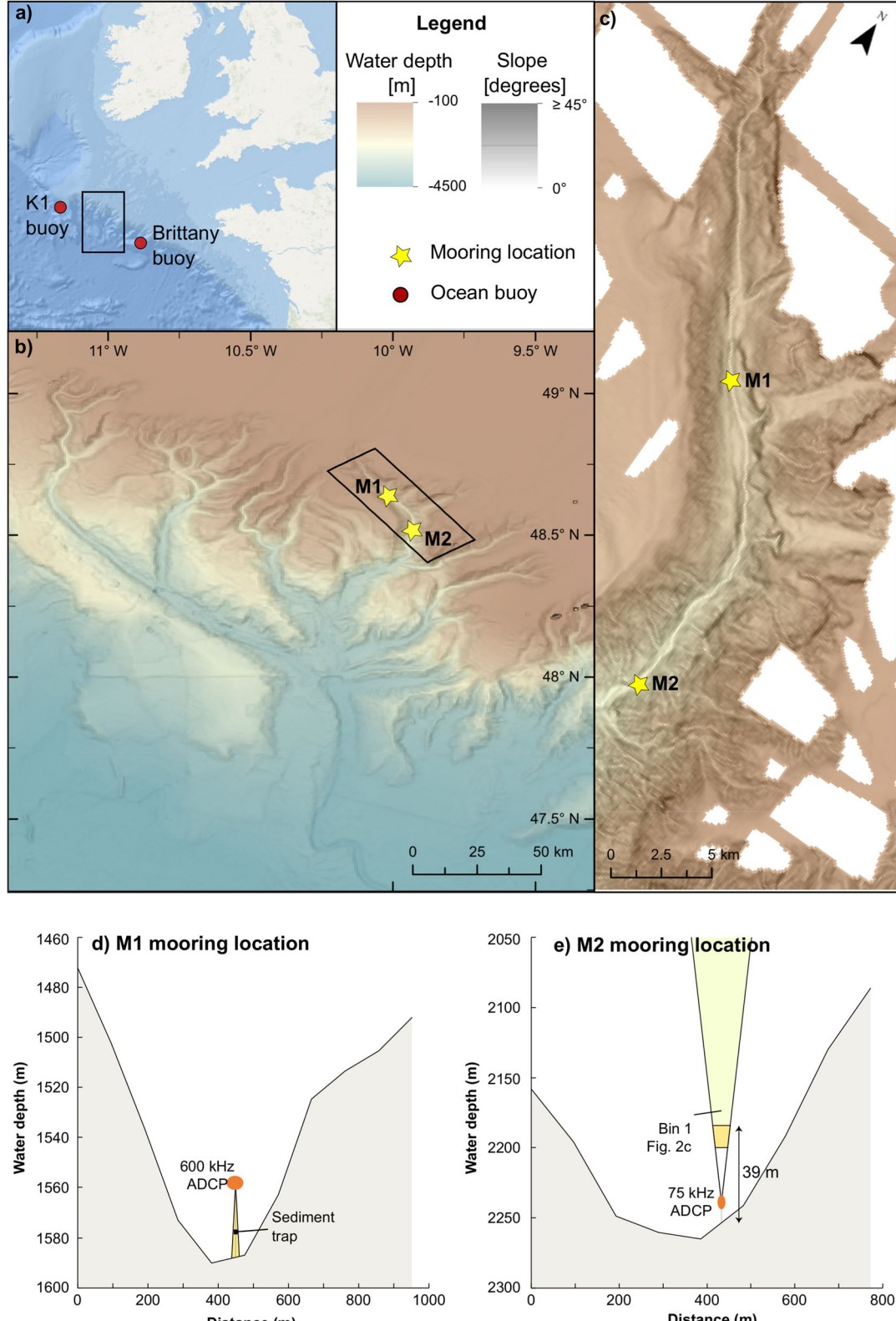

**Fig. 1 Location of the Whittard Canyon and the instrument array. a** Location of the Whittard Canyon, which is separated from the closest coastline by 300 km of continental shelf, and the location of the closest two meteorological ocean buoys. **b** Overview of the Whittard Canyon. Bathymetric metadata and Digital Terrain Model data products in panel a are derived from Esri Ocean Basemap which is based on contributions from Garmin, GEBCO and NOAA NGDC (https://www.arcgis.com/apps/mapviewer/index.html?webmap=67ab7f7c535c4687b6518e6d2343e8a2) and panel **b** are derived from the open access EMODnet Bathymetry Consortium (2020): EMODnet Digital Bathymetry (DTM) (https://doi.org/10.12770/bb6a87dd-e579-4036-abe1-e649cea9881a). Coloured bathymetric elevation is semi-transparently overlain on the greyscale slope map. **c** Location of the two moorings in the eastern branch of the Whittard Canyon. **d**, **e** Schematic figures of the layout of the ADCPs on M1 and M2 and their position. The profiles are taken perpendicular to the canyon thalweg.

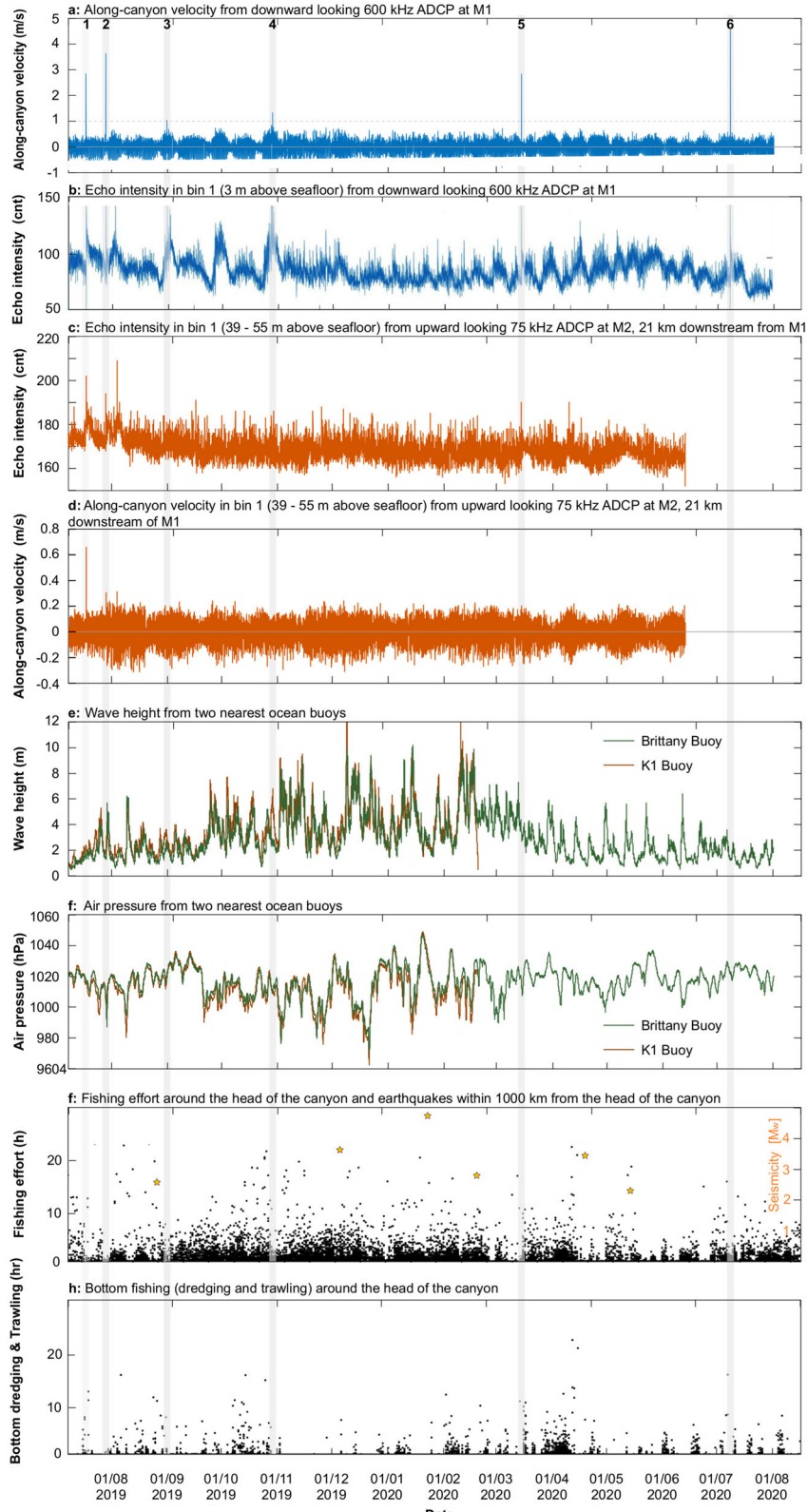

reaches of the system[27,30,35]. However, this is the first field study to document that these flows can occur on such a short recurrence (i.e. sub-annual frequency), and velocities exceed previous measurements in the Whittard Canyon by an order of magnitude. Down-canyon flows of <0.8 ms$^{-1}$ were previously recorded from benthic landers at 1400 and 4200 m water depth, but could not

always be confidently differentiated from the background effects of internal tides in the canyon, which regularly attain near-bed velocities of ≥0.5 ms$^{-1}$ [13,30]. Perhaps most surprising, is that the observed frequency (6 in a year) and velocity (up to 5–8 ms$^{-1}$) of flows in Whittard Canyon is on a par with highly active land-attached deep-sea submarine canyons (Table 2), such as the river-

**Fig. 2 Time series of monitoring data recorded at moorings and ocean buoys that document background environmental conditions and the timing of the six recorded turbidity currents.** Overview of the measurements at M1 and M2, and at the two meteorological ocean buoys: **a** maximum velocity recorded at M1. Note six peaks ($>1\,ms^{-1}$) in maximum velocity, which characterise the turbidity currents (highlighted by grey transparent fill). Positive velocities refer to down-canyon flow. **b** echo intensity at the lowermost bin (3-4 m above seafloor) of the ADCP at M1 expressed in counts (cnt). **c** echo intensity and (**d**) velocity at the lowermost bin (39–55 m above seafloor) of the ADCP at M2 (Fig. 1e). Note how turbidity currents 1, 2, and 5, coincide with enhanced velocity or backscatter at M2. Some peaks in echo intensity do not coincide with high velocity down-canyon flows and are thought instead to relate to sediment re-suspension by internal tides as shown in previous studies[47,48]. Turbidity currents 3 and 4 cannot be correlated between M1 and M2. **e**, **f** wave height and air pressure from the two meteorological ocean buoys (Fig. 1a). Note storms are not a consistent trigger that can explain the occurrence of the turbidity currents. Similarly, no clear link appears to exist with g) earthquakes within 1000 km of the canyon (orange stars), nor fishing activity, including bottom dredging and trawling around the canyon head (**h**).

connected Congo Canyon (6 flows in 4 months at 2 km water depth; $1–2.4\,ms^{-1}$ [17]) and the littoral-fed Monterey Canyon (15 flows in 18 months, of which only 3 reached 1.9 km water depth; $1–8\,ms^{-1}$ [19]). Indeed, the velocities and frequency reported here are some of the highest yet directly recorded from turbidity currents worldwide (Table 2).

**A major trigger is not required for turbidity currents, nor is a consistent trigger in effect.** Based on inference from NE Atlantic land-attached canyons (e.g. Nazaré Canyon), storms were thought to be responsible for triggering previously reported turbidity currents in Whittard Canyon[13,30]. While Flow 2 coincided with a storm (local minimum in air pressure and maximum in wave height and wind speed), many of the other storms during the monitoring period, including more vigorous events, did not correlate with the occurrence of turbidity currents (Fig. 2). Surprisingly, no turbidity currents were recorded during the European winter storm season, when wave heights and air pressures experience the largest excursions[36] (November–March; Fig. 2). Analysis of other triggers proposed for turbidity currents in submarine canyons, including earthquakes, surface tidal and internal tidal currents, and seafloor disturbance by fishing, showed no consistent finding. Five earthquakes of magnitude 2 and greater occurred within 1000 km of the canyon head, but none coincided with any of the flow timings, and no larger ($>M_w$ 6) earthquakes occurred within 2000 km during the monitoring period (Table S1). Turbidity currents occur during both spring and neap tides and are not correlated with any particular phase of the semidiurnal surface tidal cycle. Flow 1 occurs when the surface tidal flow is down-canyon, but Flows 2, 4 and 6 occur when the flow is up-canyon (Fig. 4). Near-bed currents caused by the trapping of internal tides occur along the canyon[13,30,37,38]. However, turbidity currents occur during periods of both low and high internal tide magnitude, as well as during both down-canyon and up-canyon phases of near-bed internal tidal flow (Fig. 4). Fishing occurs daily around the head of Whittard Canyon (Fig. 2f) and there is no obvious connection to turbidity current inception when considering all types of fishing. When considering only fishing activities that directly disturb the seafloor, some flows (e.g. Flows 1,4,5) appear to coincide with periods of heightened dredge fishing and bottom trawling (Fig. 4g). However, this relationship is far from equivocal, as flows did not occur on other days that had much higher bottom fishing intensity.

It appears that turbidity currents in Whittard Canyon do not require a major trigger, and instead likely occur following a period of preconditioning (i.e. from sustained or sudden sediment supply), at which point a number of minor perturbations are capable of initiating a flow. This adds to growing evidence of seasonally-clustered activity that was first documented in land-attached canyons, where preconditioning during and after periods of heightened sediment supply governs turbidity current timing and frequency, rather than requiring external triggers. For river-fed canyons, seasonal pulses in river discharge are the primary

control on sediment supply, while for land-attached canyons fed by long-shore drift, heightened wave energy during the winter storm season explains turbidity current activity[26]. In Whittard Canyon, however, the storm season is absent of turbidity currents and instead the more meteorologically-quiescent summer months are more active. Sediment transport on the Celtic Shelf is complex, as is the topography of the dendritic branches of the Whittard Canyon, which exerts a strong control on hydrodynamic processes such as internal tides[13,30,37,39]. We suggest that the 'switch on' outside of the storm season may be explained by seasonal variability of cross-shelf transport on the Celtic Margin and from the adjacent Bay of Biscay (to the SE), wherein sediment transport toward the canyon head is enhanced during summer months[39–42]. We conclude that the lack of consistency in a trigger, and the seasonal clustering of flows, result from this combined complexity of spatiotemporally-variable hydrodynamic processes and sediment supply, which may also be further complicated by anthropogenic disturbances such as shelf-edge and deep sea fishing.

**Underestimation of contemporary particulate transport in land-detached canyons.** This recognition of frequent highstand turbidity current activity in land-detached canyons is important, as more than 75% of the >9000 submarine canyons worldwide are land-detached (Fig. S3). We cannot infer that all these land-detached canyons will be similarly active. Canyons of the Antarctic margin, for instance, are formed and maintained by dense cascades of cold water[43]. However, analysis of a global database reveals at least 10% of the world's canyons have a very similar setting to Whittard Canyon; separated >100 km from shore by a broad shelf, where sediment stocks have accumulated since the Last Glacial Maximum, typically occurring on passive margins[1,2,11]. We infer a 50% increase in the number of canyons worldwide that may potentially feature active turbidity currents in the present-day highstand, including some of the largest systems on Earth[20,44]. An additional $n = 1162$ such land-detached canyons can therefore be added as potentially active systems, to the existing $n = 2104$ land-attached canyons that efficiently convey particulate matter from shelf to deep sea (Fig. S3). While such canyons may not connect directly to terrestrial sources of modern organic carbon, these systems can still be effective conveyors of fresh labile organic carbon to the deep sea. In the Whittard Canyon, phytoplankton blooms can generate elevated quantities of phytodetritus that is rich in organic carbon[30]. As these blooms occur in the spring and summer[30] (i.e. the same period within which we observe most frequent turbidity currents), it is conceivable that the powerful flows we observe play a role in the down-canyon transfer of fresh organic material, as well as anthropogenic material such as the discarded fishing gear found wrapped around our the anchor chain of mooring M1 and M2 (Fig. S1). To further quantify particulate fluxes, and constrain the potential for their activity, it is important to better characterise the nature and quantity of sediment transport on the continental

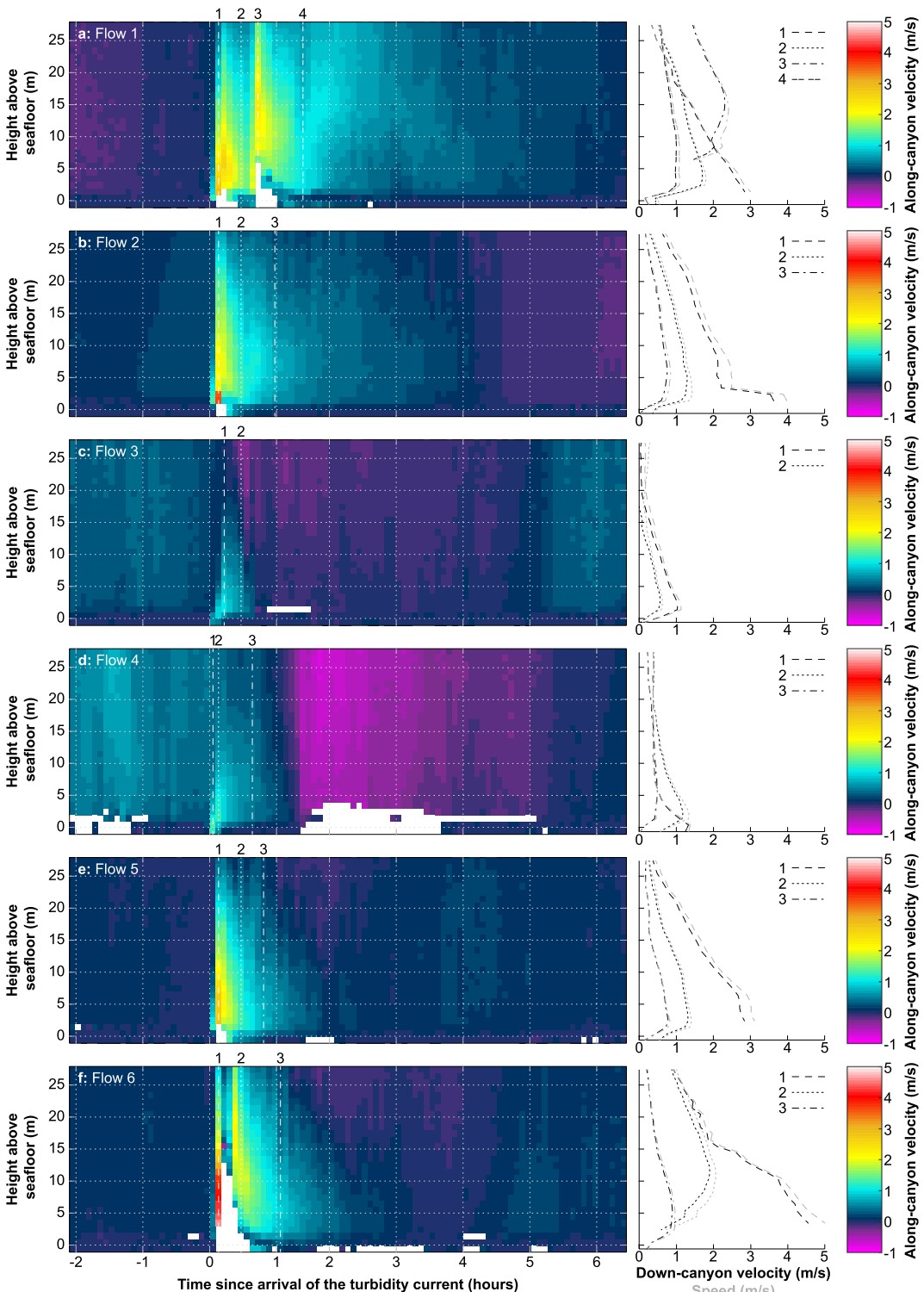

**Fig. 3 Velocity time series (left) and velocity profiles (right) of the six turbidity currents recorded in the Whittard Canyon during the monitoring period.** The highest velocities occur at the start of the flow and are located towards the base. The ADCP signal at this fastest basal part of the flow is partially attenuated at the start of flows 1, 2, 5, 6, shown as blanked (white) data. We present vertical profiles (right of down-canyon flow velocity (black dashed lines) and the maximum measured speed in any direction (grey dashed lines) at various time steps throughout these flows that correspond to numbered intervals on the time series panels (left).

shelf adjacent to these canyons[26,29,34], the energy and rate of cross-shelf transport relative to the canyon head[39], and the relative role of human activities, such as bottom fishing, that can markedly influence sediment fluxes at the shelf-scale[41]. Future monitoring efforts should thus not focus solely on land-attached

canyons and must include land-detached canyons, many of which host well-documented carbon, nutrient, pollution, and biodiversity hotspots and intersect routes for new seafloor cables that are vulnerable to the impacts of turbidity currents[8,12,45,46].

**Table 1 Measured arrival time, velocity, duration, thickness, and calculated transit velocities for turbidity currents recorded in the Whittard Canyon during the monitoring period at moorings M1 and M2.**

| Flow | Date | Mooring M1 (1591 m water depth) | | | | Mooring M2 (2259 m water depth) | |
|------|------|--------------------|----------------------------------------|-------------------|-------------------|------------------|------------------------|
| | | Arrival time (UTC) | Maximum down-canyon velocity (ms⁻¹) | Duration (hours) | Flow thickness (m) | Arrival time (UTC) | Transit velocity (ms⁻¹) |
| 1 | 17/07/2019 | 21:45 | 2.8 | 4.2 | >30 | 22:44 | 3.0–5.9 |
| 2 | 28/07/2019 | 21:00 | 3.6 | 3.3 | >30 | 21:44 | 3.5–8.0 |
| 3 | 31/08/2019 | 17:00 | 1.0 | 0.7 | 20 | Not recognised | — |
| 4 | 29/10/2019 | 03:35 | 1.3 | 1.1 | 15 | Not recognised | — |
| 5 | 14/03/2020 | 16:30 | 2.8 | 1.7 | >30 | 17:44 | 2.7–4.7 |
| 6 | 08/07/2020 | 04:45 | 4.6 | 2.9 | >30 | Outside survey period | — |

**Table 2 Comparison of the activity of turbidity currents in land-attached submarine canyons with that in the land-detached Whittard Canyon.**

| Canyon type | Location | Velocity (ms⁻¹) | Frequency of flows | Duration (hours) | Water depths of mooring (km) |
|-------------|----------|-----------------|--------------------|------------------|------------------------------|
| Land-detached | Whittard Canyon (this study) | 1.1–8.0 | 6 yr⁻¹ | 0.7–4.2 | 1.5–2 |
| Land-attached – long-shore drift | Monterey Canyon, California[19] | 1–8.1 | 10 yr⁻¹ (15 flows between Oct 15 and Apr 17) | 0.5–6 | 0.2–1.9 |
| Land-attached – long-shore drift | Eel Canyon, California[50] | Up to 0.5 | 10 s yr⁻¹ | N/A - Not reported | 0.1 |
| Land-attached – long-shore drift | Hueneme and Mugu Canyons, California[31] | ~2 | 6 yr⁻¹ and 4 yr⁻¹ respectively (3 and 2 flows between Sep 07 and Mar 08) | ~12 | 0.2 |
| Land-attached – long-shore drift | Nazaré Canyon[51,52] | >0.5 | 4 yr⁻¹ (9 flows between Oct 02 and Dec 04) | Few hours | 1.6–4.3 |
| Land-attached river-fed | Congo Canyon, W Africa[17] | 1–2.4 | At least 6 yr⁻¹ (6 flows between Dec 09 and Mar 2010) | 125–242 | 2 |
| Land-attached river-fed | Gaoping/Kaoping Canyon, Taiwan[53,54] | 3.7–20 | 6 yr⁻¹ (23 flows between May 13 and Oct 16) | N/A - Not reported | 2.1 |
| Land-attached river-fed | Var Canyon, Mediterranean[55] | 0.2–0.7 | 3 yr⁻¹ (7 flows between Oct 05 and Mar 08) | 8–22 | 1.2–2.4 |

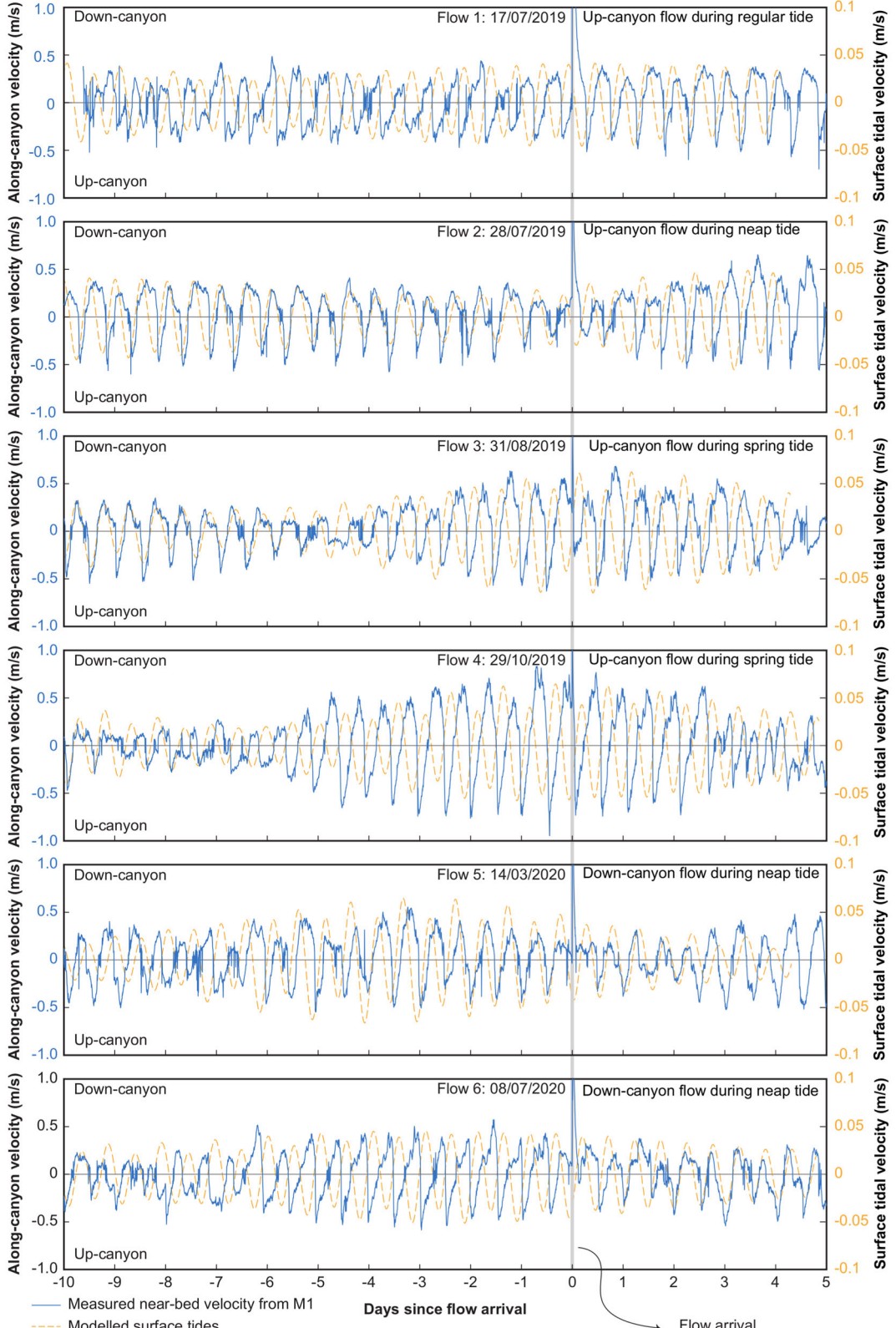

**Fig. 4 Near-bed current velocities (blue solid line; ms⁻¹) and surface tidal velocities**[25] **(orange dashed line; ms⁻¹) for ten days prior to and five days after the six major turbidity currents recorded in the Whittard Canyon.** No consistent link can be made to the modelled surface tide as turbidity current occurrence is not correlated with any particular phase of the semidiurnal surface tidal cycle (i.e. down-canyon or up-canyon tidal flow) or phase of the spring-neap cycle. Similarly, turbidity current occurrence is not correlated with direction or magnitude of measured near-bed flows (assumed to be dominated by the internal tide).

In contradiction to previous models, we report that the land-detached Whittard Canyon features frequent (sub-annual) and powerful (up to 5–8 ms$^{-1}$) turbidity currents that reach at least 2 km water depth. No consistent trigger explains these flows, with seasonal changes in off-shelf sediment transport implicated as the cause for their seasonally-variable frequency. The rate and magnitude of these flows is surprisingly similar to large active land-attached canyons, which are considered to be efficient conduits of contemporary particulate transfer to the deep sea. We conclude that the deep-sea transport of nutrients, pollutants and organic carbon via land-detached submarine canyons in the present-day may have been dramatically underestimated and propose that future monitoring efforts in land-detached submarine canyons are needed to constrain global budgets and fluxes.

## Methods

We analyse new detailed monitoring data recorded over 12 months from the eastern branch of the land-detached Whittard Canyon in the NE Atlantic (Fig. 1). We deployed high frequency (75–600 kHz) Acoustic Doppler Current Profilers (ADCPs) on two deep-water moorings (1591 m and 2259 m water depth, at a distance of up to 47 km from the canyon head; Fig. 1) to record sediment transport events within the canyon at high temporal (5 min) and vertical (up to 1 m) resolution. We access data from two of the UK Met Office's network of offshore buoys to provide hourly meteorological data (Figs. 1 and 2).

**Mooring M1.** The mooring containing the main instrument to record turbidity currents (M1) was installed 26 km (measured along canyon) from the head of the canyon at 1591 m water depth at 48.626° N, 10.004° W and recorded from 19 July 2019 to 1 August 2020. This main instrument, a downward-looking 600 kHz ADCP, was positioned 30 m above the seafloor and had a blanking distance of 2 m (Fig. 1d). This ADCP was programmed to record an ensemble of 75 pings every 5 min and operated at 1 m bin sizes, effectively making a measurement of the average velocity every 5 min at 1 m intervals through the water column.

**Mooring M2.** A second mooring (M2) was located 21 km downstream from M1 at 2259 m water depth at 48.490° N, 9.936° W. This mooring contained an upward-looking 75 kHz ADCP at 14 m above the seafloor. The ADCP was set to record ensembles consisting of 22 pings every hour, operated in 16 m vertical bins, and had a blanking distance of 24 m, effectively making the first measurement bin span 39–57 m above seafloor, with a range of 300 m. The ADCP on this mooring was recorded from 5 June 2019 to 13 June 2020. The low position and upward-facing nature of this lower frequency ADCP and bin-size precluded the identification of turbidity currents <39 m thick.

**Identification of turbidity currents and differentiating them from internal tides at M1.** The downward-looking ADCP at M1 records both turbidity currents and internal tides. We can distinguish between turbidity currents and internal tides in Whittard Canyon as turbidity currents have a short-lived (<1 h) down-canyon velocity maximum (>0.5 ms$^{-1}$) just above the bed, which then rapidly decreases higher in the water column. This velocity signature is accompanied by a sudden change in acoustic backscatter (Fig. 1b), often attenuating ADCP measurements in the lowermost part of the flows. This characteristic velocity and acoustic backscatter is similar to previously recorded turbidity currents in land-attached canyons[17,19]. Internal tides have well-defined, known periods, and typically lower peak velocities, and may also be accompanied by elevations in acoustic backscatter indicating that they are also episodically capable of resuspending seafloor sediment[47,48] (Fig. 1b). Coherent internal tide signals are usually observed through the majority of the water column, not just near the seafloor[49]. Despite this, it is possible that there may be turbidity currents <0.5 ms$^{-1}$ that cannot be reliably distinguished from the internal tides[13]. Therefore here, we necessarily focus on turbidity currents that are discernably higher in velocity than the internal tides.

**Identifying turbidity currents at M2 and calculating transit velocities between mooring locations.** Both the spatial and temporal resolution of the ADCP at M2 are too coarse, and the first measurement bin is too high above the seafloor, that detecting turbidity currents solely based on the data from M2 is challenging. We therefore searched for increases in down-canyon flow velocity and acoustic backscatter shortly after turbidity current arrival times at M1. Transit velocities of turbidity currents were determined by dividing the along-canyon distance between M1 and M2, by the difference in arrival time of the turbidity currents at each mooring location. These transit velocities are presented here as a range, since the ADCP has ensemble intervals, and therefore a temporal resolution, of 1 h.

**Comparison of turbidity currents in Whittard Canyon with those in land-attached canyons.** Several other studies have been used to compare the turbidity currents in the Whittard Canyon with those in land-attached canyons (Table 2). The search was limited to ocean-scale canyons, and hence fjord, lake, and prodelta studies have been excluded.

**Sediment trap analysis.** A McLane Parflux 7 g (21 cup) sediment trap was attached to mooring M1 at 10 m above the seafloor, which includes a funnel that overlies a mechanical carousel that turns every 18 days to present a new 500 ml sampling bottle. When recovered, the funnel was observed to be mostly infilled, and only the first sampling bottle contained any material; the carousel mechanism having apparently jammed, likely as a result of sediment grains being drawn into the interface between fixed and moving mechanical components. As a result this sedimentation event must have occurred within the first 18 days and we presume that this was due to the first turbidity current. Five sediment samples were taken from within the funnel and one from the sampling bottle to determine grain size. Three aliquots of each sample were dispersed in 30 ml 0.05% sodium hexametaphosphate solution and shaken for 24 h. The dispersed aliquots were analyzed using a Malvern Mastersizer 2000 using laser diffraction of suspended sediment grains (10,000 counts) to measure grain size distributions. Grain size distributions were measured three times per aliquot. Aliquots showed intra-sample variations of <3%. Standard reference materials showed intra-sample variations of up to 3% and accuracy toward reference values of 1.5%. Samples of sediment from the funnel show a consistent unimodal grain size distribution with a median grain diameter (D$_{50}$) of 121 μm, while the underlying bottle (that represents the first accumulation of sediment; presumably at the arrival of the flow front) is coarser (D$_{50}$ = 154 μm), but also with a unimodal distribution.

**Analysis of potential triggers.** We now outline the datasets and methods used for the investigation of potential triggers for the turbidity currents observed in Whittard Canyon. Meteorological data from two ocean buoys managed by the UK MetOffice were used to analyse potential triggers of the turbidity currents. These buoys were located at 47.558° N, 8.465° W (Brittany Buoy) and at 48.747° N, 12.454°W (K1 buoy). The Brittany Buoy was active throughout the entire survey period, but the K1 Buoy was only active up to 19 February 2020. These buoys recorded a range of meteorological data at one-hour intervals including wind speed and direction, wave height and period, and atmospheric pressure. Earthquake records were obtained through the earthquake catalogue of the U.S. Geological Survey (https://earthquake.usgs.gov/). The database was searched for earthquakes of magnitudes >2 within 1000 km of the head of Whittard Canyon, and >6 within 2000 km. To investigate the potential influence of tides, we extracted tidal velocities at the head of the Whittard Canyon using the European Shelf solution of the TPXO global model of ocean tides[25]. This model allows us to determine the surface tide amplitude and phase away from fixed tide gauge stations. The modelled surface tide and measured internal tide are in phase with respect to the 14-day spring-neap cycle, but on semidiurnal (12-h) timescales there is an incoherent phase lag between along-canyon surface tide velocity and internal tide velocity (Fig. 4). This phase lag, and the fact that the measured near-bed velocities are a factor of 10 larger than the modelled surface tide velocities, indicates that the observed semidiurnal oscillations in near-bed velocity are indeed predominately an internal tide signal. The incoherent phase lag results from the slower and variable phase speed of internal tides - dependent on water depth, stratification and background flow, which can retard or advance peak internal tide velocity at the mooring site. We assess whether internal tides may play a role in triggering turbidity currents, specifically if up- or down-canyon flows are more prone to turbidity current events, by identifying along-canyon velocity at M1, from the closest measurement to the seafloor, immediately before the arrival of each event (Fig. 4). Fishing activity data were downloaded from Global Fishing Watch (https://globalfishingwatch.org/) and formatted in estimated daily fishing effort (in hours) per 0.01° × 0.01° grids. The cumulative daily fishing effort for all grid cells around the head of the eastern branch of the Whittard Canyon (48.5°–49° N and 9.8°–10.4° W) was first calculated (Fig. 2f). We then extracted only the fishing operations that included those that disturb the seafloor (i.e. dredge fishing and trawling; Fig. 2g; Fig. S4).

**Determining comparable land-detached submarine canyons that may be similarly active.** Since Whittard Canyon experiences frequent, contemporary turbidity current activity, we wish to understand whether other land-detached canyons are similar. To do this, we make use of previous geomorphological mapping of submarine canyons worldwide[1,2,44] that includes n = 9477 submarine canyons and has formed the basis of other global studies of these features[11]. We first define land-attached canyons as canyons that are <25 km from the shoreline (n = 2104). We then search for canyons similar to Whittard Canyon, which we define here as canyons > 100 km from the shore, and <50 km from a major shelf (N = 1162). Major shelves are defined as shelves >5000 km². Our analysis shows that around 10–12% of all canyons in the world are detached from land by a major shelf. This is a 45–55% increase of potentially active canyons, from just land-attached canyons. Most new potentially active canyons are located along the

eastern margins of North America and South America, southern and western Australia and the Celtic Margin (Fig. S3).

## Data availability

The current monitoring data recorded from the 600 kHz ADCP on the M1 mooring are available via the British Oceanographic Data Centre at: https://www.bodc.ac.uk/resources/inventories/cruise_inventory/report/17695/. Further information and data pertaining to the mooring design are available in the NERC cruise report (http://nora.nerc.ac.uk/id/eprint/525366). Current monitoring data from the 75 kHz ADCP on the M2 mooring are available via the NIOZ Data Archive System at https://dataverse.nioz.nl/dataset.xhtml?persistentId=doi:10.25850/nioz/7b.b.7c. Bathymetric data for the Whittard Canyon are available from the EMODnet bathymetry portal at https://portal.emodnet-bathymetry.eu/. Meterological monitoring data from the K1 (https://www.metoffice.gov.uk/weather/specialist-forecasts/coast-and-sea/observations/162029) and Brittany buoy (https://www.metoffice.gov.uk/weather/specialist-forecasts/coast-and-sea/observations/162163) can be requested under open access for research purposes from MetOffice DataPoint (https://www.metoffice.gov.uk/services/data/datapoint). Global Fishing Watch AIS-based fishing effort and vessel presence datasets can be requested under open access for research purposes from https://globalfishingwatch.org/. The global mapping of submarine canyons is available at https://www.bluehabitats.org/. Earthquake data are available open access from the United States Geological Survey at https://earthquake.usgs.gov/.

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

## Acknowledgements

M.S.H., A.G., B.J.B., J.H., C.P., V.A.I.H., and M.A.C. acknowledge funding from Natural Environment Research Council (NERC) National Capability Programme (NE/R015953/1) "Climate Linked Atlantic Sector Science". F.M. is supported by the Innovational Research Incentives Scheme of the Netherlands Organisation for Scientific Research (NWO-VIDI Grant no. 0.16.161.360). We thank the Captain and crew of RRS Discovery on Cruises DY103 and DY116, Nick Rundle, Paul Provost and Billy Platt for their efforts in preparing, deploying and recovering the M1 mooring. We thank the Captain and crew of RV Pelagia, as well as the NIOZ technicians, for their essential assistance during the deployment and recovery of the M2 mooring. These efforts are particularly noteworthy due to the logistical challenges of mooring recovery during the early phases of the COVID-19 pandemic. Ship time in relation to the M2 mooring was provided by the Royal Netherlands Institute for Sea Research. We thank the staff of the British Ocean Sediment Core Research Facility for assistance with analysis of sediment trap samples. We thank Michele Rebesco and an anonymous reviewer for their constructive and helpful reviews.

## Author contributions

The experiment was initially conceived by M.A.C., V.A.I.H., and F.M. Mooring deployment and recovery were performed by A.R.G., B.J.B., C.P., and F.M. Internal and surface tidal analysis was performed by R.H., M.S.H. and M.A.C. M.S.H. led the analysis of ADCP, meteorological and earthquake data analysis in conjunction with M.A.C., I.A.K., F.M., V.A.I.H. and R.A.H. E.L.S. analysed fishing data with contributions from B.J.B. and V.A.I.H. J.H. performed grain size analysis on sediment trap samples. M.S.H. performed the global analysis of submarine canyons. The writing of the paper was led by M.A.C. and M.S.H., with contributions from all authors. Figures were drafted by M.A.C., M.S.H. and E.L.S.

## Competing interests

The authors declare no competing interests.
