## [Peer Review File · Nature Communications]

Challenging the highstand-dormant paradigm for land-detached submarine canyonsReviewers' Comments:

Reviewer #1:

Remarks to the Author:

It was really a pleasure to read this paper. It is very well written and accurately prepared.

The methodology seems sound to me, but I cannot comment much on it as in my work I mostly deal with sediments already deposited and normally rely on physical oceanography colleagues for the analysis of ADCP data.

The experiment seems to me properly conducted and the results (frequency and speed in a land-detached canyon) are significant by themselves. But the implications for global budgets of sediment, nutrients and pollutants are the most noteworthy results.

Given that monitoring has focused on land-attached canyons, this work is highly novel and provides an excellent complement to established literature on submarine canyons.

Of course, more monitoring efforts including land-detached canyons is needed in the future, but the evidence provided here seems enough to support the conclusions and implications of this work. I don't see any major flaw in the interpretation.

Best regards,
Michele Rebesco

Reviewer #2:

Remarks to the Author:

The manuscript "Challenging the highstand-dormant paradigm for land-detached submarine canyons" by Heijnen, M.S. et al. (Ref. NCOMMS-22-11062-T) provides evidence of frequent (sub-annual) and powerful turbidity currents affecting the Whittard Canyon, which is located about 300 km far from the coastline and disconnected from major terrestrial sediment inputs in present-day highstand conditions. Despite it has been considered currently inactive with respect to turbidity currents, through the analysis of ADCP data from two moorings at 1591 and 2259 m depth, the authors show the occurrence of several sedimentary flow events, whose frequency and speed are comparable to those observed in other large active shelf-indenting canyons.

The findings presented are valuable and suggest that even canyons indenting large shelves and located hundreds of km far from the coastline may be currently more active than previously thought, potentially opening a new scenario on the role of canyons in particulate transfer to the deep sea. The manuscript is well written and fitting for the journal. I therefore recommend publication of this work after minor improvements.

In the results, the description of the instruments (lines 80-91) is too detailed and I think it would be more suitable for the Methods section. I would suggest to include basic information about the method used to conduct the study (location and depth of the two ADCP, type of recorded data...) at the end of the introduction, while leaving the more detailed information about the characteristics of the equipment (i.e. frequency, blanking distance, recording time intervals...) in the methods. The results section could then start with the actual findings of this study.

In the methods section, the authors describe how they distinguished turbidity currents from internal tides, using velocity and backscatter information. However, the backscatter data from M1 are not shown. Also, in figure 2, I see several peaks of high echo intensity at M2 (even the highest recorded) not related to turbidity currents. How do you explain these peaks? I think this should be clarified somewhere in the manuscript.

In the methods section, the meteorological data analysis is reported as a separate subsection, but I think it could be included in "analysis of potential triggers" along with the analysis of earthquakes,

tides, and fishing effort.

Other minor suggestions and typographical corrections are:

Line 27: land-detached

Line 49: sea-level

Line 82: Fig. 1

Line 86: in methods, the blanking distance reported is 24 m. Please check

Line 91: Figs. 1 & 2

Line 91: Figs. 2 & 3

Line 98: range from 2.7 to 8.0 ms⁻¹

Lines 141-142: delete the comma between near-bed and currents

Line 148: add reference to Fig. 4g

Lines 152-153: check punctuation. I would replace the semicolon with comma and delete the comma at line 153

Line 212: missing space (1591 m – 2259 m)

Line 228: 21 km downstream from M1

Line 244: through

Line 560 – “can are available” – please delete “can”

Line 274-275: I think this sentence is more suitable for the result section.

Line 296 and 310: add line breaks

Line 326-328: see previous comment

Figures:

Figure 1. Legend is also for panel a, where the location of the buoys is shown. I cannot see the slope gradient map (it seems that panel c has the same depth color-scale as in b). In Panel d, it is shown the microCAT location but it is not reported in the figure legend. I guess it also indicates the location of the sediment trap, please specify.

Figure 2: why the backscatter data is shown only for M2?

Figure 4: the left axis in the first plot should be blue

Supplementary material:

Line 44: check doubled verb

Figure S4: the legend for the fishing intensity (hours of trawling as green tones) is missing. Why only M1 is shown?

Responses to Reviewers

We thank the two reviewers and the editor for their constructive feedback on our manuscript. We were very glad Reviewer 1 found the paper to be “*very well written and accurately prepared*” and commented that “*this work is highly novel and provides an excellent complement to established literature on submarine canyons*”, and Reviewer 2 recognised “*the findings presented are valuable*”, “*potentially opening a new scenario on the role of canyons in particulate transfer to the deep sea*”, and also agreed that “*the manuscript is well written and fitting for the journal*”. Neither reviewer raised any major queries; hence below we simply provide a response to their minor comments. We have attached a tracked changes and clean version of the manuscript in order that our revisions can be identified clearly.

In addition to the reviewer suggestions, we have made the following revisions:

- We have now uploaded the M2 mooring data to an open access archive that will be made live once the paper is published (as the data are currently under embargo until publication); hence we have added new text to the Acknowledgements section as follows: “Current monitoring data from the 75 kHz ADCP on the M2 mooring are currently under embargo stored in the NIOZ Data Archive System (<https://dataverse.nioz.nl/dataverse/doi>) and will be available once accepted for publication at doi: 10.25850/nioz/7b.b.7c.”
- We have also added two further photographs to illustrate the fishing gear snagged around the M2 mooring (Fig S1).
- Inclusion of acknowledgement text recognizing the efforts of the captain and crew involved in the deployment and recovery of the moorings, which was particularly challenging in the early phases of the COVID-19 pandemic.

The following outlines our point-by-point responses to the reviewer comments:

Response to Reviewer #1:

It was really a pleasure to read this paper. It is very well written and accurately prepared. The methodology seems sound to me, but I cannot comment much on it as in my work I mostly deal with sediments already deposited and normally rely on physical oceanography colleagues for the analysis of ADCP data. The experiment seems to me properly conducted and the results (frequency and speed in a land-detached canyon) are significant by themselves. But the implications for global budgets of sediment, nutrients and pollutants are the most noteworthy results. Given that monitoring has focused on land-attached canyons, this work is highly novel and provides an excellent complement to established literature on submarine canyons. Of course, more monitoring efforts including land-detached canyons is needed in the future, but the evidence provided here seems enough to support the conclusions and implications of this work. I don't see any major flaw in the interpretation.

We thank the reviewer for their positive feedback and assessment of our paper. We are glad that they agree with our conclusions, particularly recognizing the implications for global deep sea particulate transfer and that this monitoring studies fills a knowledge gap.

We acknowledge their comment that more monitoring efforts in land-detached canyons will be needed in the future, as supported by our concluding statement “Future monitoring efforts should thus not focus solely on land-attached canyons and must include land-detached canyons...”.

Response to Reviewer #2

The manuscript “Challenging the highstand-dormant paradigm for land-detached submarine canyons” by Heijnen, M.S. et al. (Ref. NCOMMS-22-11062-T) provides evidence of frequent (sub-annual) and powerful turbidity currents affecting the Whittard Canyon, which is located about 300 km far from the coastline and disconnected from major terrestrial sediment inputs in present-day

highstand conditions. Despite it has been considered currently inactive with respect to turbidity currents, through the analysis of ADCP data from two moorings at 1591 and 2259 m depth, the authors show the occurrence of several sedimentary flow events, whose frequency and speed are comparable to those observed in other large active shelf-indenting canyons.

The findings presented are valuable and suggest that even canyons indenting large shelves and located hundreds of km far from the coastline may be currently more active than previously thought, potentially opening a new scenario on the role of canyons in particulate transfer to the deep sea. The manuscript is well written and fitting for the journal. I therefore recommend publication of this work after minor improvements.

We thank the reviewer for this positive assessment and for their careful review. We are pleased that they also clearly grasp the key conclusions and the importance of our findings.

In the results, the description of the instruments (lines 80-91) is too detailed and I think it would be more suitable for the Methods section. I would suggest to include basic information about the method used to conduct the study (location and depth of the two ADCP, type of recorded data...) at the end of the introduction, while leaving the more detailed information about the characteristics of the equipment (i.e. frequency, blanking distance, recording time intervals...) in the methods. The results section could then start with the actual findings of this study.

We agree that this text is not truly describing “Results”; hence we agree that moving it to the Introduction section is more appropriate. As a consequence, we have moved the text. We prefer to keep the high level information on resolution and blanking distance as this is important context for understanding the results (particularly as the sensor packages on the two moorings are not identical). As a result, we have moved it to the Introduction but it is our preference to leave the text unmodified and we note that Reviewer 1 did not have any issue with this information being retained in the main text of the manuscript.

In the methods section, the authors describe how they distinguished turbidity currents from internal tides, using velocity and backscatter information. However, the backscatter data from M1 are not shown. Also, in figure 2, I see several peaks of high echo intensity at M2 (even the highest recorded) not related to turbidity currents. How do you explain these peaks? I think this should be clarified somewhere in the manuscript.

We have now included backscatter data for mooring M1 in Figure 2, which were not shown previously. Peaks in backscatter that are not attributable to turbidity currents likely relate to (re)suspension of sediment by internal tides as has previously been shown by Hall et al. (2017) and Haalboom et al. (2021).

We have added text to the manuscript to clarify this point as suggested by the reviewer, including new text in the caption of Figure 2 “Some peaks in echo intensity do not coincide with high velocity down-canyon flows and are thought instead to relate to sediment re-suspension by internal tides” and at Line 282 (in tracked changes version) in the Methods: “Internal tides have well defined, known periods, and typically lower peak velocities, and may also be accompanied by elevations in acoustic backscatter indicating that they are also episodically capable of resuspending seafloor sediment^{54,55} (Fig. 1b).”

Hall, R.A., Aslam, T. and Huvenne, V.A., 2017. Partly standing internal tides in a dendritic submarine canyon observed by an ocean glider. *Deep Sea Research Part I: Oceanographic Research Papers*, 126, pp.73-84.

Haalboom, S., de Stigter, H., Duineveld, G., van Haren, H., Reichart, G.J. and Mienis, F., 2021. Suspended particulate matter in a submarine canyon (Whittard Canyon, Bay of Biscay, NE Atlantic Ocean): Assessment of commonly used instruments to record turbidity. *Marine Geology*, 434, p.106439.

In the methods section, the meteorological data analysis is reported as a separate subsection, but I think it could be included in “analysis of potential triggers” along with the analysis of earthquakes, tides, and fishing effort.

As suggested, we have moved the “meteorological data analysis” text to the “analysis of potential triggers” section of the Methods.

Other minor suggestions and typographical corrections

Line 27: land-detached

Typo corrected as suggested

Line 49: sea-level

Typo corrected as suggested

Line 82: Fig. 1

Typo corrected as suggested

Line 86: in methods, the blanking distance reported is 24 m. Please check

Corrected to 24 m here.

Line 91: Figs. 1 & 2

Typo corrected as suggested

Line 91: Figs. 2 & 3

Typo corrected as suggested

Line 98: range from 2.7 to 8.0 ms⁻¹

Typo corrected as suggested

Lines 141-142: delete the comma between near-bed and currents

Comma deleted

Line 148: add reference to Fig. 4g

Reference added as suggested.

Lines 152-153: check punctuation. I would replace the semicolon with comma and delete the comma at line 153

Semicolon replaced and comma deleted as suggested.

Line 212: missing space (1591 m – 2259 m)

Space added before units as suggested.

Line 228: 21 km downstream from M1

Modified as suggested.

Line 244: through

Modified as suggested.

Line 560 – “can are available” – please delete “can”

Corrected as suggested.

Line 274-275: I think this sentence is more suitable for the result section.

An equivalent sentence exists in the Results section so we have removed this statement here.

Line 296 and 310: add line breaks

Line 326-328: see previous comment

Comments on Figures:

Figure 1. Legend is also for panel a, where the location of the buoys is shown. I cannot see the slope gradient map (it seems that panel c has the same depth color-scale as in b). In Panel d, it is shown the microCAT location but it is not reported in the figure legend. I guess it also indicates the location of the sediment trap, please specify.

Legend modified accordingly. The slope gradient map is shown behind the semi-transparent bathymetric elevation rendering. New text has been added to the caption of Fig. 1 to clarify: "Coloured bathymetric elevation is semi-transparently overlain on the greyscale slope map". The location of the microCAT has been removed and instead the location of the sediment trap is now shown.

Figure 2: why the backscatter data is shown only for M2?

We have now added backscatter data for M1. See earlier comments.

Figure 4: the left axis in the first plot should be blue

This is corrected as suggested.

Comments on supplementary material:

Line 44: check doubled verb

Caption modified as suggested, i.e. "are defined" deleted.

Figure S4: the legend for the fishing intensity (hours of trawling as green tones) is missing. Why only M1 is shown?

Mooring M2 now added to this figure.

Reviewers' Comments:

Reviewer #2:

Remarks to the Author:

The authors have carefully addressed all the minor comments on previous version of the manuscript. I have no further comments and I think the paper and suitable for publication.

Thank you for the opportunity to revise this very interesting paper.

Response to Reviewers

We thank the reviewer and editor for their review. Below we respond to the reviewer comments (which are reproduced verbatim as requested).

"Reviewer #2 (Remarks to the Author):

The authors have carefully addressed all the minor comments on previous version of the manuscript. I have no further comments and I think the paper and suitable for publication.

Thank you for the opportunity to revise this very interesting paper".

We are pleased that the reviewer is happy with our revisions.